# Screening coffee genotypes for brown eye spot resistance in Brazil

**Juliana Barros Ramos**[ID]¹, **Mario Lucio Vilela de Resende**¹, **Deila Magna dos Santos Botelho**¹, **Renata Cristina Martins Pereira**[ID]¹, **Tharyn Reichel**[ID]¹, **André Augusto Ferreira Balieiro**[ID]¹, **Gustavo Pucci Botega**[ID]², **Juliana Costa de Rezende Abrahão**[ID]³*

1 Dept. de Fitopatologia, Universidade Federal de Lavras, Lavras, MG, Brasil, 2 Dept. de Genética e Melhoramento de Plantas, Universidade Federal de Lavras, Lavras, MG, Brasil, 3 Empresa de Pesquisa Agropecuária de Minas Gerais, Campus Universitário, Lavras, MG, Brasil

\* julianacosta@epamig.br

## Abstract

Several researchers have attempted to develop coffee plants that are resistant to brown eye spot (BES); however, no coffee cultivars are resistant to the disease. In the present study, a blend of strains from *Cercospora coffeicola* was inoculated into 19 Brazilian commercial cultivars and 41 accessions from the Germplasm Collection of Minas Gerais to evaluate the genetic resistance ability within the population and select superior genotypes for the breeding program. After predicting the genotypic values of the estudied material, the evaluations number necessary for selecting genotypes with accuracy and efficiency was determined based on the data of severity to BES. The action of defense mechanisms plant was also investigated by assessing the levels of total soluble phenolic compounds and soluble lignin in contrasting genotypes for disease susceptibility. Based on the results, the accession MG 1207 Sumatra, had an intrinsic genetic capacity to maintain low levels of severity to BES. The genotype MG 1207 Sumatra can substantially contribute to the development of new cultivars, which may lead to the reduced use of pesticides. According to the accuracy and efficiency results obtained, four evaluations BES severity are sufficient to achieve accuracy, providing expressive genetic gains. Finally, the levels of lignin and phenolic compounds were not found to be associated with the resistance of coffee genotypes to BES.

## Introduction

Coffee is a crop of great economic importance in many countries including Brazil, the largest producer and exporter of coffee in the world [1]. The country is internationally recognized for its ability to produce coffee in large volumes and at competitive prices, but, recently, it also stands out as a prestigious origin of specialty coffees. Brazil is also the world's second largest consumer of the beverage, surpassed only by the United States of America [2, 3].

The occurrence of diseases is most important limitations in the coffee production, especially brown eye spot (BES), which is considered a disease economic importance for coffee growers in Brazil and othercontries coffee producers as Colombia, Porto Rico, Costa Rica, El-Salvador and Honduras [4]. The disease caused by fungus *Cercospora coffeicola* Berk. & Cooke

**Data Availability Statement:** Data are fully available without restriction: Ramos, Juliana et al. (2022), Screening coffee genotypes for brown eye

spot resistance in Brazil, Dryad, Dataset, https://doi.org/10.5061/dryad.4j0zpc8d6.

**Funding:** The authors wish to thank National Counsel of Technological and Scientific Development- CNPq, the Brazilian Consortium Coffee Research and Development, the National Institute of Coffee Science and Technology (INCT Café/CNPq), and the Foundation for Research Support of the State of Minas Gerais (FAPEMIG) for their financial support. The funders had no role in study design, data collection and analysis, decision to publish, or preparation of the manuscript.

**Competing interests:** The authors have declared that no competing interests exist.

has host *Coffea* species as *Coffea arabica*, *Coffea canephora*, *Coffea eugenioides*, *Coffea liberica* and *Coffea racemosa* (Echandi, 1959). In the absence of management measures losse may reach up to 30% [5]. The BES is characterized by leaf symptoms round necrotic lesions, with a brown outer ring and a gray-white center area [6, 7]. Further, BES causes defoliation, stimulates maturation, besides intensifying pulp adherence to the endocarp, which makes it difficult to carry out de-pulping affecting the quality of the produced coffee [6].

Increases in the productive potential in coffee areas can lead to nutritional imbalance, resulting in a higher susceptibility to this disease [7]. Current BES management has been commonly accomplished via the application of chemical products coupled with an accurate water supply and plant nutrition [8]. In Brazil, increases in the incidence of BES is related with the expansion of coffee fields to other regions with different environmental conditions.

To date, as the physiology of plant responses underlying genetic resistance is quite complex, few studies have been carried out on BES resistance in coffee species [9–11]. In this context, Dell'Acqua et al. [9] and Patricio et al. [10] used the phenotypic data from BES severity evaluation under greenhouse conditions to carry out selection based on a test of means. These researchers considered the studied genotypes as fixed effects, without estimating the genetic variance and heritability of the phenotypic variation. This strategy may not be effective in crop breeding programs that aim to develop materials resistant to BES as it does not assess the potential effects of selection on the studied characters.

Plants have a set of defense mechanisms to defend themselves against pathogen attacks, which can either be constitutively structural or biochemical [12]. Among the induced mechanisms, the production of phenolic compounds and lignin is commonly observed during defense reactions, which are produced via the metabolism of phenylpropanoids [13], thereby providing higher resistance to the plant cell wall during pathogen attack. Understanding how infection process for *C. coffeicola* affects the biochemical aspects of plant cells can shed light on a further mechanism to control BES.

This study aimed to select superior *Coffea arabica* genotypes for resistance to BES and quantify the total soluble phenolic compounds and soluble lignin in contrasting genotypes for disease susceptibility. The possibility of selecting coffee plants with higher resistance to this disease is extremely important; however, the success of this strategy depends on the existence of genetic variation within studied characteristics, and a high heritability rate.

## Material and methods

### BES severity

The experiment was carried out under greenhouse conditions at the Department of Phytopathology of the Universidade Federal de Lavras—UFLA, Brazil. Coffee seedlings of 19 commercial cultivars and 41 accessions from the Germplasm collection of the Agricultural Research Corporation of the State of Minas Gerais (EPAMIG) in Patrocínio, MG, were evaluated to determine their resistance to BES (S1 Table). The accessions were selected according to their characteristics of yield, drink quality, and/or resistance to other diseases of economic importance.

Seeds were sown in 5 L plastic trays containing autoclaved sand. The germination chamber was adjusted to 30˚C and 80% relative humidity. The seedlings were transplanted into punched-black polyethylene pots (0.11 x 0.20 m) at the phase of cotyledons. The substrate consisted of 300 L of cattle manure and 700 L of soil mix extracted from the 0.4 to 0.8 m layer of Dystrophic Red Latosol and fertilized with 5 kg of simple superphosphate and 500 g of potassium chloride.

The seedlings were inoculated at the stage of three pairs of true leaves. The inoculum consisted of a blend of different isolates of *C. coffeicola* obtained from coffee leaves with symptoms of BES collected in the municipalities of Marechal Floriano (ES), Cachoeirinha, Ervália, Lavras, and Patrocínio (MG). Different isolates were employed to include a higher variability of the fungus [6, 11].

The sporulation of the isolates was performed as described by Souza et al. [14], with adaptations. Eight mycelial discs (6 mm in diameter) were removed from the colony borders (on day 15 of growth) of different isolates of *C. coffeicola*. Discs were macerated in 400 μL of sterile distilled water. The macerated mycelium from each isolate was placed in Erlenmeyer flasks containing 20 mL of liquid V8 culture medium (100 mL of V8 in 900mL of distilled water) under shaking at 100 rpm for 12 d at room temperature. The liquid containing the mycelium was transferred to plates containing a water-agar medium. The plates were kept in a BOD incubator (Bio-Oxygen Demand), with a photoperiod of 12 h at 25°C. After culture medium dehydration (approximately 5 days of incubation), 10 mL of sterile water was added to each plate and the conidia were removed with a Drigalski spatula. The suspension was filtered with sterile gauze to remove residues and the conidia were subsequently quantified in a Neubauer chamber. The suspension used for inoculation was adjusted to $5 \times 10^4$ conidia·mL$^{-1}$ and sprayed on the abaxial side of the leaves of all seedlings using a manual sprayer. Thereafter, the seedlings were placed in a humid chamber for 72 h.

Temperature and relative humidity data were collected over the experimental period with a Datalogger HT-500, Instrutherm®. Weekly assessments of disease severity were performed on the first two pairs of true leaves throughout the five weeks, starting from the onset of symptoms (approximately 15 d after inoculation). The severity of BES in different coffee genotypes was quantified using a diagrammatic scale with six classes of the proportion in the infected area by BES [15]: class 1: 0.1–3.0%; class 2: 3.1–6.0%; class 3: 6.1–12.0%; class 4: 12.1–18.0%, class 5: 18.01–30.0%, and class 6: 30.1–50.0%. The experiment was repeated twice. The distribution of the phenotypic segregation was evaluated in each genotype within the six classes described. Based on this assessment, the resistance level was determined: class 1- resistant (R); class 2- partially resistant (PR); class 3- moderately susceptible (MS); class 4- susceptible (S); classes 5 and 6- highly susceptible (AS).

The severity dates were used to calculated thea area under disease progress curve (AUDPC), as previously proposed by Shaner and Finney [16].

$$AUDPC = \sum_{i=1}^{n-1} \left( \frac{si + si + 1}{2} \right)(ti + 1 - ti) \tag{1}$$

where AUDPC is area under disease progress curve; Si is disease severity in the time of evaluation; and ti is the time of evaluation i.

## Total soluble phenolic compounds and soluble lignin

Both total soluble phenolic compounds and soluble lignin levels were quantified in the leaves of MG 1207 accession (presenting low disease severity in the present study) and MG 0291 and Catuaí Vermelho IAC 144 genotypes (presenting high disease severity). The Catuaí Vermelho IAC 144 genotype was used as a control of susceptibility, according to Patricio et al. [10] and Botelho et al. [11].

The samples consisted of 2nd and 3rd pairs of fully expanded leaves, collected at 24, 120, 240, 480, and 720 h after inoculation (hai) of *C. coffeicola*. Samples of non-inoculated plants with the pathogen were also collected at 24 and 720 h to confirm that the inoculation influences the levels of total soluble phenolic compounds and soluble lignin. After collection, the samples were immediately stored in liquid nitrogen and keep an ultra-freezer until sample processing.

The macerated samples were lyophilized and approximately 30 mg of the material was homogenized in 80% methanol. The solution was centrifuged at room temperature for 5 min at 14000 rpm. The supernatant and the precipitate were used to quantify total soluble phenolic compounds and soluble lignin, respectively.

The levels of total soluble phenolic compounds were determined as described by Spanos and Wrolstad [17], with modifications. The supernatant was homogenized with 0.25 N Folin-Ciocalteu reagent, 1 M Na$_2$CO$_3$, and distilled water. The reaction was standardized in 200 μL and quantified using a spectrophotometer at 725 nm. Based on the standard curve of chlorogenic acid, the levels of total soluble phenolic compounds were calculated.

Lignin was quantified as proposed by Doster and Bostock [18]. The precipitate was homogenized in 80% methanol and centrifuged as described for phenolic compounds. The contents were evaporated in an oven at 45˚C overnight and mixed with thioglycolic acid and 2 M HCl (ratio 1:10) in a water bath at 100˚C for 4 h. After centrifugation and solubilization in 0.5 M NaOH, the supernatant was homogenized with HCl P.A., and kept at 4˚C for 4 h before centrifugation. The precipitate was homogenized in 0.5 M NaOH. A 200-μL aliquot of this solution was used for the reaction, and the absorbance was assessed in a Power Wave XS microplate spectrophotometer (Biotek®) at 280 nm. Based on a standard curve of lignin, the soluble lignin content was subsequently estimated. The quantification of total soluble phenolic compounds and soluble lignin contents was performed in triplicate.

## Statistical analysis

The experiment BES severity were conducted in a randomized complete block design, with 60 treatments (genotypes), eight replicates, and five evaluations. The experimental plot comprised two coffee plants. A mixed model analysis was accomplished using the Selegen REML/BLUP software [19]. The following equation was used: $y = Xm+Zg+Tp+e$, where $y$ is the data vector; $m$ is scalar referring to the general average of fixed effect; $g$ is the vector of random genetic effects; $p$ is the vector of the random-effects of blocks within replications and experiment; and $e$ is the vector of random errors. X, Z, and T are incidence matrices of referred effects. The variance components were subjected to the likelihood ratio test at 5% probability.

The experiment of quantification of total soluble phenolic compounds and soluble lignin in coffee genotypes, were conducted in a randomized complete block design with three replicates and two plants per plot. Analysis of variance was conducted in a 3 x 5 factorial scheme, with three evaluated genotypes and five collection times (24, 120, 240, 480, and 720 h) after inoculation (hai) of *C. coffeicola*. The data were subjected to analysis of variance and the means were compared by Tukey's test at 5% probability.

## Results

### Genotype selection

The analysis based on mixed models provided a simultaneous estimation of both individual heritability and plot repeatability (Fig 1, S1 Table). The heritability and genetic variation coefficients were significant, thereby indicating the possibility of selection. The selection among families was high (98%) and should be recommended. The coefficient of permanent effect determination showed a low magnitude (‹0.00), revealing that variations in the environment among measurements do not affect such responses. The overall average of the experiment was over 14.50% of severity (S1 Table).

The studied population presented a high genotypic variability for the severity of BES. The predicted additive values were between 2.37% (accession MG 1207) and 32.02% (Catuaí

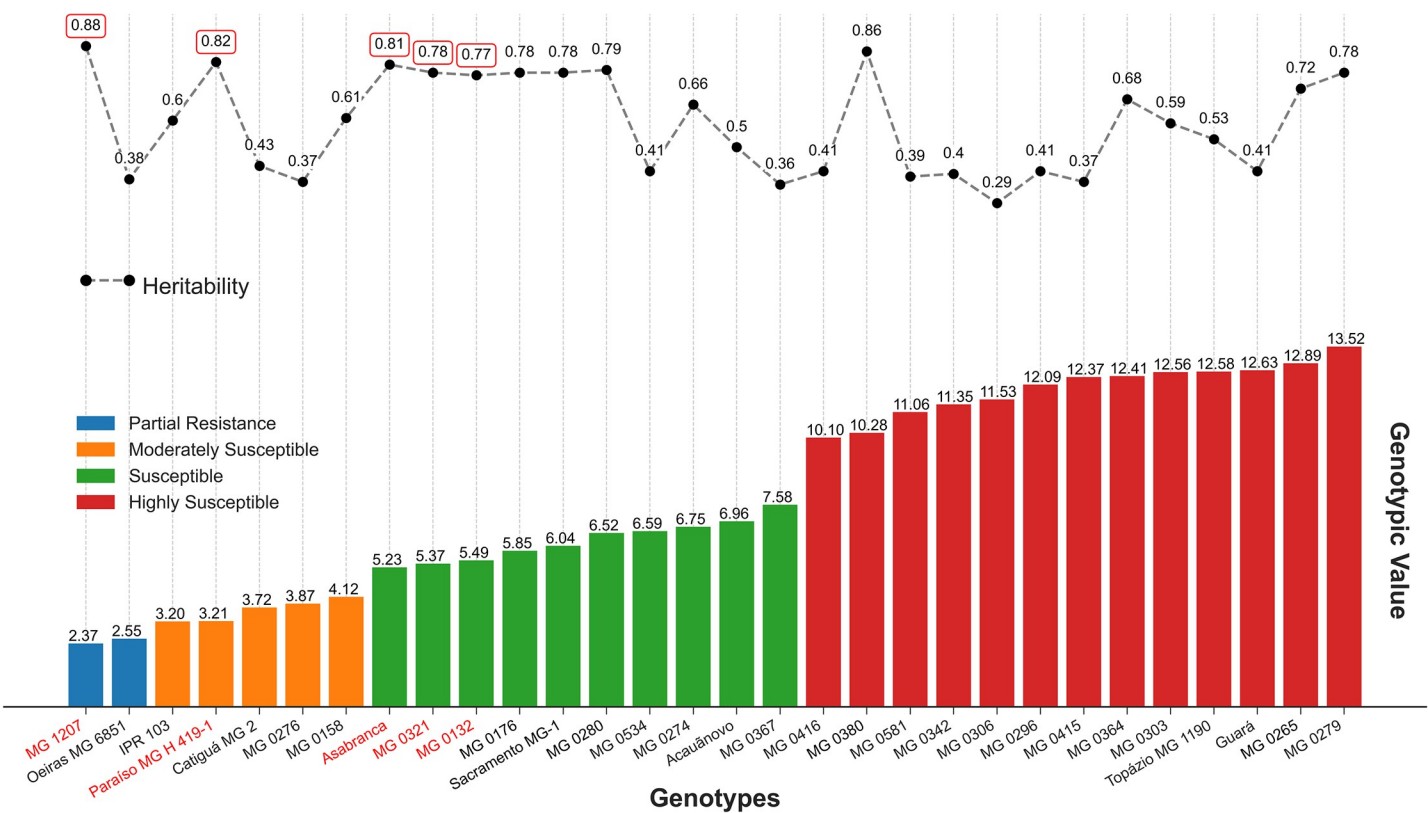

**Fig 1. Acessions classification based on phenotypic mean, genotypic values (u + g), genotype heritability, calculated on BES severity in coffee seedlings.** The genotypes in red color represents heritability presented heritability (H) of high magnitude (higher than 70%).

Vermelho IAC 144); while values for the individual heritability were between 29 and 88% among 60 genotypes (S1 Table).

Based on the analysis of frequency-severity, the genotypes were mainly clustered into 4, 5, and 6 classes (Table 1), and were thus considered susceptible or highly susceptible. The accessions, MG 1207 Sumatra and Oeiras MG 6851, displayed injured leaf area by approximately 3.1 to 6.0%, and were classified as partially resistant (class 2). The accessions MG 0276 and MG 0158, and the cultivars, Catiguá MG2, Paraíso MG H 419–1, and IPR 103, were considered moderately susceptible (class 3). None of the genotypes studied were clustered in the first class (0 to 3% of injured area).

Among the studied population, approximately 38% presented heritability (H) of high magnitude (higher than 70%), namely MG 1207, MG 0321, MG 0132, MG 0176, MG 0280, MG 0380, MG 0265, MG 0279, MG 0324, MG 0267, MG 0723, MG 0333, MG 0134, MG 0179, MG 0282, MG 0663, and MG 0291 accessions, besides the cultivars Paraíso MG H 419–1, Asabranca, Araponga MG1, Sacramento MG1 and Catucaí Amarelo 2SL and Siriema AS 1. Within

**Table 1. Average heritability, accuracy, and efficiency based on different number of measurements for the severity of brown eye spot.**

| Measurements | 1 | 2 | 3 | 4 | 5 | 6 | 7 | 8 | 9 | 10 |
|---|---|---|---|---|---|---|---|---|---|---|
| Heritability | 0.90 | 0.95 | 0.97 | 0.97 | 0.98 | 0.98 | 0.98 | 0.99 | 0.99 | 0.99 |
| Accuracy | 0.95 | 0.97 | 0.98 | 0.99 | 0.99 | 0.99 | 0.99 | 0.99 | 0.99 | 0.99 |
| Efficiency | 1.00 | 1.14 | 1.20 | 1.23 | 1.25 | 1.27 | 1.28 | 1.29 | 1.30 | 1.30 |

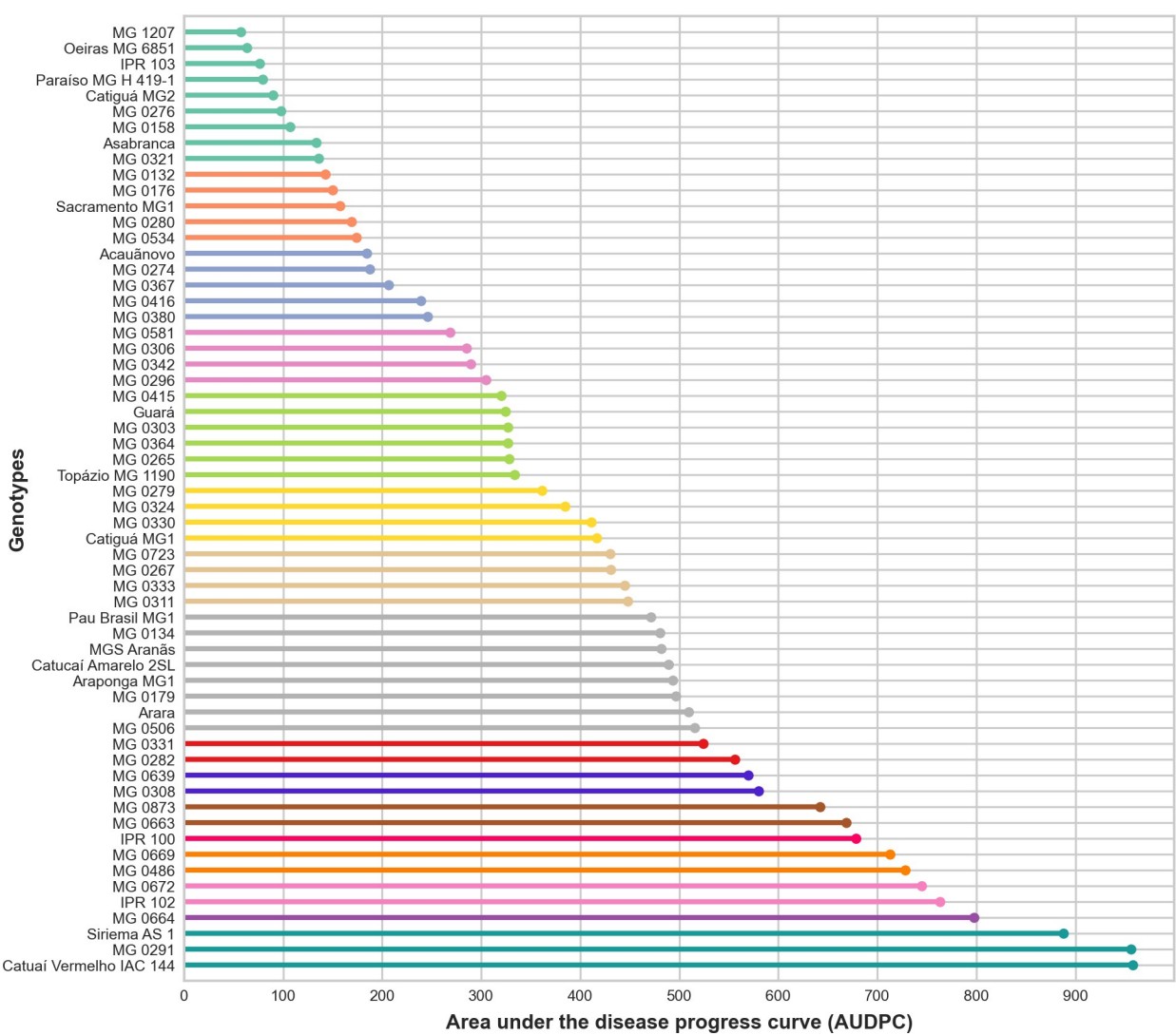

**Fig 2. Area under disease progress curve (AUDPC).** *Bars with the same color do not differ from each other by the Scott-Knott test ($p \leq 0.05$).

this population, there was a high variation in the severity of BES, with a phenotypic value ranging from 4.6 to 50.

We selected five genotypes with the highest predicted additive values (between 2.37 and 5.49) and the highest magnitudes of individual heritability (H) (between 77 and 88%), such as the MG 1207, MG 0321, and MG 0132 accessions, and the commercial cultivars, Paraiso MG H 419–1 and Asabranca, which presented low magnitude for their genotypic value to reduce severity, suggesting further probabilities of genetic progress through these evaluations. The selection of five such genotypes resulted in a remarkable selection gain of 70.16% to reduce the severity of BES.

Based on the phenotypic data, the outputs from AUDPC severity for BES were significantly different among the genotypes. Otherwise, by considering the heritability parameter, the selection via AUDPC highlighted MG 1207, Oeiras MG 6851, IPR 103, Paraíso MG H 419–1, and Catiguá MG 2 genotypes, which showed the lowest disease progress during the evaluated timeframe (Fig 2). Genotype MG 0291, Catuaí Vermelho IAC 144, and Siriema AS 1 were the most

susceptible among them, with AUDPC values of 958.04, 956.24, and 888.08, respectively. By comparing the genotypes that demonstrated the highest and lowest AUDPC, a difference in disease severity of 901.71% was obtained.

Accuracy can be improved using more rigorous analysis methods, such as by increasing the number of severity measurements per plant. In this context, Table 1 presents the accuracies that would be achieved using a higher number of measurements. Considering the estimated individual repeatability, the adoption of four measurements leads to accuracies around 99%.

## Total soluble phenolic compounds and soluble lignin content in coffee genotypes

Based on the results of brown eye spot severity, accessions from Sumatra group were selected for quantification of total soluble phenolic compounds and soluble lignin. They were MG 1207, as low severity, and MG 0291, which presented a high severity, being thus placed alongside the control (Catuaí Vermelho IAC 144).

There were no significant differences in total soluble phenolic compounds and soluble lignin levels among studied genotypes. On the other hand, there was a significant interaction between time of collection for both the contents of phenolic compounds (Fig 3) and soluble lignin (Fig 4).

Phenolic compounds content showed variation among the three genotypes over time (Fig 3), with the lowest levels at 480 hai, presenting mean values between 5.76 and 6.90 μL/mg. The highest levels of phenolic compounds were observed at 24 hai in Catuaí Vermelho IAC 144 (7.40 μL/mg), and 720 hai in MG 1207 (7.58 μL/mg) and MG 0291 (7.28 μL/mg). The highest lignin contents were found in genotypes at 720 hai, which ranged from 10.79 to 12.17 μg/mg. Otherwise, the lowest levels of this compound were observed in MG 1207 (6.51 μg/mg) at 240 hai, and MG 0291 (5.82 μg/mg) and Catuaí Vermelho IAC 144 (6.29 μg/mg) at 24 hai.

By analyzing the soluble lignin contents at different collection times, the MG 0291 accession was found to display an increased content of this compound over the timeframe (Fig 4), unlike the MG 1207 and Catuaí Vermelho IAC 144. The lignin content in such genotypes at 24 and 240 hai was lower than that observed at 120 and 480 hai, respectively. Between 480 and 720 hai, the same profile was verified in two genotypes: the last collection time was associated with a higher lignin content than previous collection times.

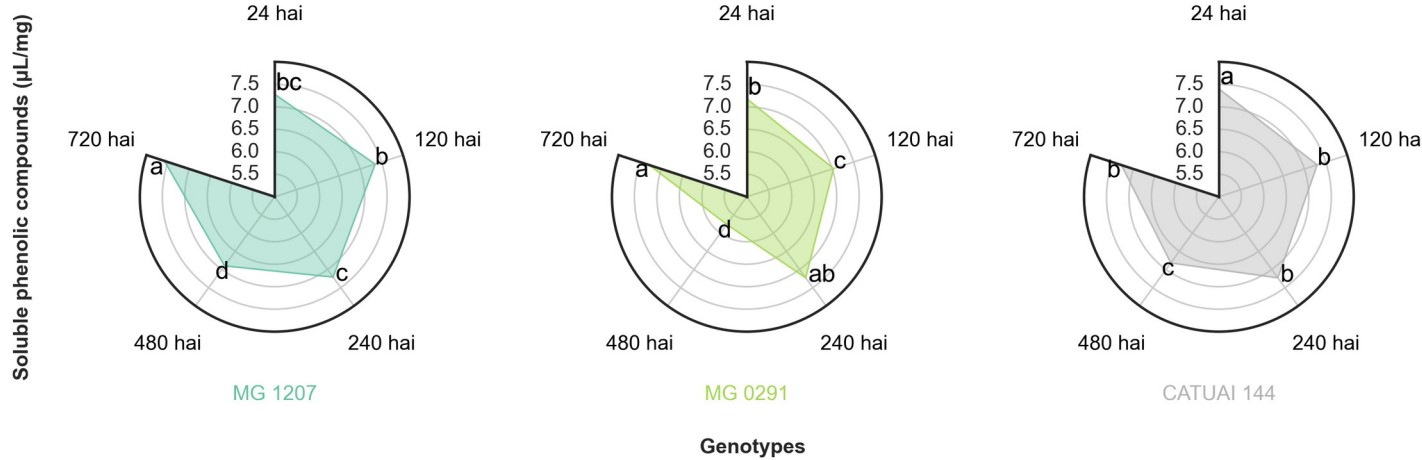

**Fig 3. Contents of total soluble phenolic compounds (μL/mg) quantified in the leaves of the genotypes MG 1207, MG 0291, and Catuaí Vermelho IAC 144 collected at 24, 120, 240, 480, and 720 h after inoculation of *C. coffeicola*.** *Averages followed by the same lowercase letter in the same genotype do not differ by the Tukey's test (p≤0.05).

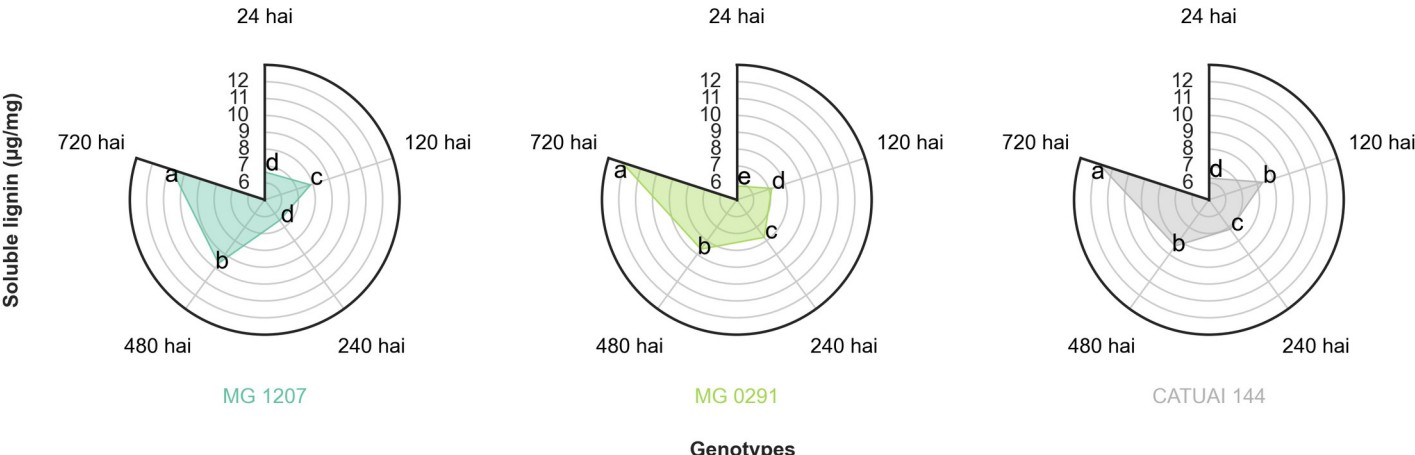

**Fig 4. Soluble lignin content (μg/mg) in the leaves of MG 1207, MG 0291, and Catuaí Vermelho IAC 144 collected at 24, 120, 240, 480, and 720 h after inoculation of *C. coffeicola*.** *Averages followed by the same lowercase letter in the same genotype do not differ by the Tukey's test (p≤0.05).

## Discussion

Due to the environmental condition that benefits BES development in most coffee growing regions, besides the lack of resistant cultivars, this disease remains a remarkable challenge faced by Brazilian coffee growers, even after 100 years of introduction of the disease in the country [20], or 40 years after the first epidemic reports [21]. In this context, several efforts have been made by researchers worldwide to develop plants that are resistant to different *Cercospora* species; however, studies on *C. coffeicola* are still incipient. The increases in the productive potential in coffee areas can lead to nutritional imbalance, resulting in a higher susceptibility to disease, which makes it difficult for breeders to select such plants in the field [7]. Thus, this study aimed to highlight strategies for the early selection of genotypes that allow genetic gains in a short timeframe, to further provide materials for the genetic breeding program to achieve resistance under field conditions.

The genotype MG 1207 Sumatra, and the cultivar, Oeiras MG 6851, were considered partially resistant, while accessions MG 0276 and MG 0158 and cultivars Catiguá MG 2, Paraíso MG H 419–1, and IPR 103 were considered moderately susceptible. All other genotypes were susceptible because they presented an injured leaf area greater than 12.0%. In the present study, no genotype was classified as resistant. The existence of partially resistant phenotypes in our study indicated that severity to BES is a quantitatively inherited character and presents an action conditioned by many genes with small individual effects; consequently, it is markedly influenced by environmental factors. In such cases, it is essential to use statistical methods based on estimates of genetic parameters, as heritability estimates enable both the development of more efficient selection strategies and prediction of the selection gain [22].

By using different approaches to analyze the outputs from severity data, only two genotypes (MG 1207 and Paraiso MG H 419–1) were found to match between both selection methods based on five genotypes selected by AUDPC (MG 1207, MG 0321, MG 0132, Paraiso MG H 419–1 and Asabranca) and five genotypes selected by the highest genotypic values and estimated heritability (MG 1207, Oeiras MG 6851, IPR 103, Paraíso MG H 419–1, and Catiguá MG 2). By estimating the heritability of each genotype, environmental effects are expected to be attenuated. Thus, the genotypes, MG 0321 and MG 0132, and Asabranca, selected by the AUDPC, presented a heritability lower than 70%, which indicates that the expression of the severity of BES in such materials has a low genetic influence, causing major responses to

environmental conditions [23]. Such genotypes are not expected to maintain their phenotypic behavior of low severity to BES over time, which makes experimental replication difficult, and may be a false positive source of resistance. Only 20 of the 60 genotypes evaluated showed high magnitude heritability (S1 Table). By considering the fixed model (Fig 2), only nine of these genotypes would have a satisfactory heritable fraction of genotypic variance—approximately 70% or more—(MG 1207, Paraiso MG H419-1, Asabranca, MG 0321, MG 0132, MG 0176, Sacramento MG1, MG 0280, MG 0380), if 20 genotypes of low severity to BES are selected. These results indicate that selection without observing the heritability of each genotype individually may not be ideal, and may lead to an inefficient method for breeding programs that seek to achieve resistance to BES.

Genetic gain is inversely proportional to the selection intensity, which was 8.3% (five genotypes) in the present study, thereby allowing higher efficiency in the subsequent steps of selection [24]. The accessions MG 1207, MG 0321, and MG 0132, and the commercial cultivars, Paraiso MG H 419–1 and Asabranca, had a lower magnitude of genotypic value to reduce the severity of BES, suggesting suitable possibilities of genetic progress in the sequence of evaluations (Table 1). Based on the severity of BES, the new predicted mean based on the selection of these five genotypes was approximately 4.33, with a remarkable selection gain of 70% to reduce the severity (Fig 1). The acessions MG 1207 and MG 0132 refer to the Sumatra group, and are parental lines of the cultivar, Mundo Novo; while accession MG 0321 is the Hibrido de Timor UFV 432–09. The cultivar, Paraiso MG H 419–1, represents the interbreeding between Catuaí Amarelo IAC 30 and the Híbrido de Timor UFV 445–46 [25]. Finally, the cultivar, Asa Branca, results from interbreeding between Sarchimor 1668 and the Mundo Novo 379–19 [26]. Therefore, these last three genotypes are resistant to the orange rust fungus of the coffee tree, *H. vastatrix.*

Accuracy is a measure closely related to the precision of selection and holds the property of establishing the evaluation reliability and predict genotype genetic value based on its magnitude [23]. In the present study, the assumption of four assessments resulted in values above or close to 99% (Table 1) by considering the estimated individual repeatability, which indicates a very high precision. Such finding suggests that there are low absolute deviations between the true genotypic values and those estimated or predicted. Such outputs make it easier to achieve expressive genetic gains. The efficiency of using four measurements compared to one measurement is approximately 1.23 or 23% for this character. These results align with the number of measurements used in Arabica coffee trees to accomplish agronomic data [27]. By doubling this number (i.e., eight measurements), efficiency only increases by 6%.

In this study, no significant differences were observed in total soluble phenolic compounds and soluble lignin levels among the inoculated and non-inoculated genotypes. Similar results are reported elsewhere on this pathosystem [28, 29]. As phenolic compounds and lignin are produced as a physical and chemical defense mechanism against pathogen attacks [30–32], the inoculated genotypes are expected to present higher levels of these metabolites than non-inoculated genotypes.

As the levels of metabolites were not found to be associated with coffee tree resistance to BES, this level may be associated with phenylpropanoid pathway behavior, from which phenolic compounds are synthesized via lignin monomers [33, 34]. By facing stress, the expression of genes involved in this pathway may be altered to produce soluble phenolic compounds, which also present functions of defense and antioxidant properties, and may not be directly involved in lignin biosynthesis [35].

The crosstalk among the biosynthesis of phenolic compounds/lignin, growth, reproduction, and plant defense are other factors that can justify the fact that the inoculated and non-inoculated genotypes of coffee plants did not differ in metabolite levels. Plant defense and growth

are negatively correlated, such as the activation of defense processes that negatively affect plant growth and reproduction [32]. Thus, in this study, plants may have triggered other physiological processes rather than the accumulation of phenolic compounds and lignin (plant defense). As lignification is a process tightly controlled by various regulatory levels, it will only occur at the appropriate time and location of lignin deposition [34, 35].

Leaf tissue necrosis is caused by the oxidation and polymerization of o-diphenols [36, 37]. In the present study, decreases in the content of phenolic compounds observed between 240 and 480 hai may be related to the use of such compounds for the synthesis of other products, such as tannins and lignans, as the time of 480 hai is close to the period of symptom manifestation of BES. However, increases in the content of phenolic compounds, verified between 480 and 720 hai, are characterized as a generic response to insects, fungi, or bacteria attacks [38]. Decreases in phenol content followed by successive increases were demonstrated elsewhere with the resistance of olive cultivars to *Verticillium dahliae* [39].

Lignin plays an important role in plant growth and development, as it promotes increased cell wall rigidity, and its metabolism is involved in the response to different biotic and abiotic stresses [40]. Accordingly, environmental stresses change the content and composition of this compound in plants, which explains the difference observed in lignin content at the different times of collection in this study.

Based on the outputs presented in this study, the accession MG 1207 Sumatra, was classified as partially resistant, which is in close agreement with the results of Botelho et al [11]. This genotype can substantially contribute to the development of a new cultivar to reduce the use of pesticides. Herein, the assumption of four assessments for the severity of BES was found to be sufficient for the parameters of accuracy and efficiency, leading to expressive genetic gains. Finally, the levels of lignin and phenolic compounds were not found to be associated with the resistance of coffee genotypes to BES.

## Supporting information

**S1 Table. Genotypes and their respective distribution of leaf lesions frequency into six classes, class 1: 0.1–3.0; class 2: 3.1–6.0; class 3: 6.1–12.0; class 4: 12.1–18.0; class 5: 18.1–30.0; class 6: 30.1–50% of the leaf surface affected by brown eye spot, phenotypic mean of the genotypes (M), classification (C) of the accessions based on the phenotypic mean, predicted additive breeding values (u + g), heritability (H) of genotypes, genetic parameters, general mean and estimated gain relative to the severity of brown eye spot in 5 selected genotypes.** PR: partially resistant, MS: moderately susceptible, S: susceptible, AS: highly susceptible. *Cultivar used as a control. CV–Catura Vermelho; HT–Híbrido de Timor $h^2_m$: heritability of genotype averages. $h^2_i$: broad-sense heritability individually for each plot, or the total genotypic effects. $c^2_{perm}$: coefficient of determination of the permanent environmental effects. $M_O$: Genotypic mean of 60 genotypes before selection. $M_S$ Mean of the five selected genotypes. SG: gain selection.
(DOCX)

## Author Contributions

**Conceptualization:** Juliana Barros Ramos, Mario Lucio Vilela de Resende.

**Data curation:** Juliana Barros Ramos, André Augusto Ferreira Balieiro.

**Formal analysis:** Juliana Barros Ramos, Gustavo Pucci Botega, Juliana Costa de Rezende Abrahão.

**Investigation:** Juliana Barros Ramos, Deila Magna dos Santos Botelho.

**Methodology:** Deila Magna dos Santos Botelho, Renata Cristina Martins Pereira.

**Software:** Gustavo Pucci Botega.

**Supervision:** Mario Lucio Vilela de Resende.

**Visualization:** Renata Cristina Martins Pereira, Tharyn Reichel.

**Writing – original draft:** Juliana Barros Ramos, Deila Magna dos Santos Botelho.

**Writing – review & editing:** Mario Lucio Vilela de Resende, Tharyn Reichel, Juliana Costa de Rezende Abrahão.

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
