## [Decision Letter · Decision Letter 0]

29 Nov 2021

PONE-D-21-31634Screening coffee genotypes for Cercospora coffeicola resistance in BrazilPLOS ONE

Dear Dr. Resende,

Thank you for submitting your manuscript to PLOS ONE. After careful consideration, we feel that it has merit but does not fully meet PLOS ONE’s publication criteria as it currently stands. Therefore, we invite you to submit a revised version of the manuscript that addresses the points raised during the review process.

We look forward to receiving your revised manuscript.

Kind regards,

Abel Chemura

Academic Editor

PLOS ONE

Journal Requirements:

Additional Editor Comments:

Comments

line 44: To avoid confusing readers, it is better to use one name for the disease and not mix between the scientific name and common name. If you choose to stick with the scientific name better to stick with that and not change between the names.

line 50-52: Revise the statement on four evaluations of severity so that it clear to readers what you are referring to.

line 60: A general statement to introduce readers to coffee as a crop of importance in Brazil and the world will be good. In addition more information on the distribution, hosts, symptoms and life cycle, impact and current management (and why current management is not sustainable) of the disease not only for Brazil but for the entire coffee sector is required here.

Line 197: Revise for clarity.

Line 286-288: “indicating that the action of defense mechanisms against the C. coffeicola attack does not differ in terms of disease 288 susceptibility”: This is not clearly understandable what you mean. Is it that the soluble phenolic compounds and soluble lignin levels are not indicators of C. coffeicola resistance?

Line 323-346: This section should be moved to the introduction and not be in Discussion.

The supplementary materials Conjuntamod69.xlsx should be stored on data repository sites and not used as SI because it contains a lot of data for this purpose.

Reviewers' comments:

Reviewer's Responses to Questions

**Comments to the Author**

1. Is the manuscript technically sound, and do the data support the conclusions?

Reviewer #1: Yes

Reviewer #2: Yes

2. Has the statistical analysis been performed appropriately and rigorously? 

Reviewer #1: Yes

Reviewer #2: Yes

3. Have the authors made all data underlying the findings in their manuscript fully available?

Reviewer #1: Yes

Reviewer #2: No

4. Is the manuscript presented in an intelligible fashion and written in standard English?

Reviewer #1: Yes

Reviewer #2: No

5. Review Comments to the Author

Reviewer #1: Research paper is based on original research work carried out by the authors, very well organized research and data was analyzed statistically and explained well in paper. paper is technically sound with supportive data with good findings of resistant lines for coffee brown eye spot disease in Brazil which help in management of the disease.

Reviewer #2: 1. The authors did not present a very good understanding of English. There is need to correct some grammatical errors in the document to meet the standards of the journal. Example is line 188 to 189; Two coffee seedling was considered an experimental unit, and many more areas need attention.

2. The authors need to clearly identify the research design utilized in the study according to the standard designs we have. The design should be called Randomized Complete Block Design rather than completely randomized block design.

3. There is need for consistency in the document when the author indicated Area Under Disease Progressions Curve and Area under curve for disease progression.

4. the graphs are lacking clear labels on the axis in order to fully understand the scales used.

6. PLOS authors have the option to publish the peer review history of their article (what does this mean?). If published, this will include your full peer review and any attached files.

Reviewer #1: No

Reviewer #2: No

---

## [Author Response · Author response to Decision Letter 0]

5 Jan 2022

To the Editorial Board of Plos one

Dear Editor,

Please find enclosed the manuscript entitled “Screening coffee genotypes for Cercospora coffeicola resistance in Brazil” PONE-D-21-31634

Journal Requirements:

Answer: References were review 

Additional Editor Comments:

Comments

line 44: To avoid confusing readers, it is better to use one name for the disease and not mix between the scientific name and common name. If you choose to stick with the scientific name better to stick with that and not change between the names.

Answer: The suggestion was answered; we standardized to use only the name of the disease

line 50-52: Revise the statement on four evaluations of severity so that it clear to readers what you are referring to.

Answer: This part of the article was rewritten (lines 43-45; 51-53)

line 60: A general statement to introduce readers to coffee as a crop of importance in Brazil and the world will be good. In addition more information on the distribution, hosts, symptoms and life cycle, impact and current management (and why current management is not sustainable) of the disease not only for Brazil but for the entire coffee sector is required here.

Answer: The introduction of the manuscript has been modified as suggested (lines 60-72).

Line 197: Revise for clarity.

Answer: AUDPC = Area under disease progress curve was corrected in the manuscript

Line 286-288: “indicating that the action of defense mechanisms against the C. coffeicola attack does not differ in terms of disease 288 susceptibility”: This is not clearly understandable what you mean. Is it that the soluble phenolic compounds and soluble lignin levels are not indicators of C. coffeicola resistance?

Answer: This part of the manuscript was corrected (line 278-286)

Line 323-346: This section should be moved to the introduction and not be in Discussion.

Answer: This section was excluded of the manuscript

The supplementary materials Conjuntamod69.xlsx should be stored on data repository sites and not used as SI because it contains a lot of data for this purpose.

Answer: We would appreciate it if you could provide us with guidance on which site we should host this data on.

Reviewer #2: 1. The authors did not present a very good understanding of English. There is need to correct some grammatical errors in the document to meet the standards of the journal. Example is line 188 to 189; Two coffee seedling was considered an experimental unit, and many more areas need attention.

Answer: The manuscript was corrected 

2. The authors need to clearly identify the research design utilized in the study according to the standard designs we have. The design should be called Randomized Complete Block Design rather than completely randomized block design.

Answer: The name design utilized in the study was corrected in the manuscript (lines 196; 199; 207)

3. There is need for consistency in the document when the author indicated Area Under Disease Progressions Curve and Area under curve for disease progression.

Answer: The correction was made (line153; 158)

4. The graphs are lacking clear labels on the axis in order to fully understand the scales used.

Answer: The graphs was remade as suggested

The figure files were uploading to the Preflight Analysis and Conversion Engine (PACE) digital diagnostic tool. All corrections had been made in the text are highlighted in red. 

Thank you for the opportunity to improve and submit the manuscript.

Juliana Costa de Rezende Abrahão, on behalf of the author

---

## [Editor Report · Decision Letter 1]

11 Jan 2022

Screening coffee genotypes for brown eye spot resistance in Brazil

PONE-D-21-31634R1

Dear Dr. Resende,

We’re pleased to inform you that your manuscript has been judged scientifically suitable for publication and will be formally accepted for publication once it meets all outstanding technical requirements. You can consider open data repositories like Dryad, Zenodo, FigShare for data archiving*. *

Kind regards,

Abel Chemura

Academic Editor

PLOS ONE
---

## [Editor Report · Acceptance letter]

20 Jan 2022

PONE-D-21-31634R1 

Screening coffee genotypes for brown eye spot resistance in Brazil 

Dear Dr. Abrahão:

I'm pleased to inform you that your manuscript has been deemed suitable for publication in PLOS ONE. Congratulations! Your manuscript is now with our production department. 

Kind regards, 

on behalf of

Dr. Abel Chemura 

Academic Editor

PLOS ONE